# A standardised differential privacy framework for epidemiological modeling with mobile phone data

Merveille Koissi Savi [1], Akash Yadav[2], Wanrong Zhang[3], Navin Vembar[4], Andrew Schroeder[2], Satchit Balsari[5], Caroline O. Buckee[6], Salil Vadhan[3], Nishant Kishore [6] *

1 Department of Medical Oncology, Dana Farber Cancer Institute, Harvard School of Medicine, Boston, Massachusetts, United States of America, 2 Direct Relief, Santa Barbara, California, United States of America, 3 Department of Computer Sciences, Harvard John A. Paulson School of Engineering & Applied Sciences, Boston, Massachusetts, United States of America, 4 Camber Systems, Washington, District of Columbia, United States of America, 5 Department of Emergency Medicine, Harvard Medical School, Boston, Massachusetts, United States of America, 6 Department of Epidemiology, Harvard TH Chan School of Public Health, Boston, Massachusetts, United States of America

* nish.kishore@gmail.com

## Abstract

During the COVID-19 pandemic, the use of mobile phone data for monitoring human mobility patterns has become increasingly common, both to study the impact of travel restrictions on population movement and epidemiological modeling. Despite the importance of these data, the use of location information to guide public policy can raise issues of privacy and ethical use. Studies have shown that simple aggregation does not protect the privacy of an individual, and there are no universal standards for aggregation that guarantee anonymity. Newer methods, such as differential privacy, can provide statistically verifiable protection against identifiability but have been largely untested as inputs for compartment models used in infectious disease epidemiology. Our study examines the application of differential privacy as an anonymisation tool in epidemiological models, studying the impact of adding quantifiable statistical noise to mobile phone-based location data on the bias of ten common epidemiological metrics. We find that many epidemiological metrics are preserved and remain close to their non-private values when the true noise state is less than 20, in a count transition matrix, which corresponds to a privacy-less parameter $\epsilon = 0.05$ per release. We show that differential privacy offers a robust approach to preserving individual privacy in mobility data while providing useful population-level insights for public health. Importantly, we have built a modular software pipeline to facilitate the replication and expansion of our framework.

## Author summary

Human mobility data has been used broadly in epidemiological population models to better understand the transmission dynamics of an epidemic, predict its future trajectory,

**Data Availability Statement:** Data and codes are available at https://github.com/crisisready/DP_Metapopulation.

**Funding:** A portion of this study was generously supported by a Trust in Science Grant awarded by the Harvard Data Science Initiative to A.S., S.B., and C.B. The funder had no role in study design, data collection and analysis, decision to publish, or preparation of the manuscript. The other authors received no specific funding for this work.

**Competing interests:** The authors have declared that no competing interests exist.

and evaluate potential interventions. The availability and use of these data inherently raises the question of how we can balance individual privacy and the statistical utility of these data. Unfortunately, there are few existing frameworks that allow us to quantify this trade-off. Here, we have developed a framework to implement a differential privacy layer on top of human mobility data which can guarantee a minimum level of privacy protection and evaluate their effects on the statistical utility of model outputs. We show that this set of models and their outputs are resilient to high levels of privacy-preserving noise and suggest a standard privacy threshold with an epsilon of 0.05. Finally, we provide a reproducible framework for public health researchers and data providers to evaluate varying levels of privacy-preserving noise in human mobility data inputs, models, and epidemiological outputs.

## Introduction

The use of private mobile phone data for various applications in public health, urban planning, and response to natural disasters has been steadily growing for more than a decade. The COVID-19 pandemic has accelerated this trend, and the use of mobility data has increased, following the need to monitor and make policy decisions related to travel restrictions and lockdowns. These data were incorporated into epidemiological models during the pandemic to monitor or forecast SARS-COV-2 transmission.

Mobility data from mobile phones allow us to quantify changes in human movement, identify how social contacts cluster, evaluate where cases come into contact with others, and predict the probability of geographic spread [1–4]. Data acquired from cell phone metadata recorded for billing purposes or from digital platforms are aggregated and shared with researchers, who can then get significant information from mobility patterns [5–7]. Such studies have been used to explain the seasonal pattern of dengue in Pakistan and rubella in Kenya, for example [5, 7]. These models are predominantly metapopulation models in which mobility data are used to determine the impact of human migration on the trajectory of infectious diseases. During the COVID-19 pandemic, the use of mobility data increased around the world, and metapopulation models were used to understand the relationship between human mobility and the spread of the epidemic, predict the dynamics of the epidemic, and estimate the effectiveness of nonpharmaceutical interventions such as lockdowns, reopenings, and social distancing, based on other work modeling the spatial dynamics of pathogens [1, 5, 6].

Despite the statistical utility of these datasets, important privacy concerns remain about the sharing of personal data, even if they are deidentified and aggregated. Standardised approaches are currently lacking for data-sharing agreements and guidelines on the appropriate ways to protect individual privacy while using mobility data for public health. As big data, the semantic web, the interconnectedness of digital technology, and the "Internet of Things" (IoT) increase the volume and velocity of data, it becomes easier to reanonymise such aggregated data [8, 9].

Several privacy frameworks have been developed to address the trade-off between privacy and utility for statistical analyses [10–13]. Amongst these frameworks, *differential privacy* (DP) has become the leading approach to balance this trade-off [14]. DP is a parameterized privacy concept, where the privacy parameter $\epsilon$ allows for a smooth trade-off between privacy and utility for statistical analyses [14]. Informally, an algorithm that is $\epsilon$-differentially private ensures that any particular output of the algorithm is at most $e^{\epsilon}$ more likely when we arbitrarily change one data entry. In DP, observations are perturbed by adding noise coming from a carefully chosen distribution [14]. A DP mechanism applied to a mobility matrix of travel between

different locations will prevent disclosing the exact number of movements and will also keep the private information of the individual (home and work location, etc.) hidden.

DP is considered the gold standard of statistical privacy, as its application can be proven to preserve privacy while quantifying the trade-off between privacy and the utility of the released statistics [15]. The trade-off between privacy and utility is important because the noisier the output, the less useful it may be for inference. Increasingly, DP is used for the public release of data sets by industries and governments such as Google [16], Apple [17], Microsoft [18], Facebook (Meta), Uber [19], and the US Census Bureau [20], but it remains unclear how DP should be used in the context of mobility data for epidemiological frameworks.

In this paper, we examine how differential privacy can be applied to infectious disease modeling and analyse the impact of different levels of privacy on the reconstruction of epidemic features through simulation. Our method is based on a previously validated epidemiological metapopulation model, and we investigate the effect of the addition of privacy-preserving noise on key epidemiological outputs of interest. We used real-world mobility data from New York State during the early stages of the COVID-19 pandemic in the United States and show that the application of differential privacy can bias certain epidemiological metrics. We propose that differential privacy offers a rigorous and quantifiable approach to safely using mobile phone data during epidemics for modeling purposes.

## Results

### Mobility data

The mobility matrices included data from August 15 to November 15, 2020, and contained a total of 812,587 transitions made between sixty-two counties of New York State, with a mean of 9,029 transitions a day. The observed daily transitions ranged from a minimum of 600, occurring in Hamilton County, to a maximum of 77,131 in Suffolk County. The maximum transition between counties occurred between Queens and Kings counties, with 5,262, whereas we counted 14 combinations of zero transitions during the selected windows. After applying DP, the absolute number of transitions was affected, but the relative rank of the intercounty routes with respect to the volume of travel remained the same. We initiated a variety of common scenarios to assess the effect of added noise on bias and variability in our epidemiological parameters of interest.

### Scenarios with initial outbreaks in large and small regions

We first address the impact of starting epidemics in large versus small counties to determine whether DP would have systematic impacts on the dynamics overall. Kings and Queens are the largest counties in New York State with an approximate population of 2 million individuals each [21]. Allegany and Essex are the smallest counties in New York state, with populations of approximately 46,000 and 37,000 individuals, respectively. In each of these counties (first the two largest, and then the two smallest), we seeded 20 infectious individuals to spark an epidemic. In the scenario with large counties, we observed epidemics that started around the 50th day and peaked around the 75th day, reaching approximately 1% of the population living in these areas. In the smaller counties, the epidemic began around the 60th day and peaked on the 150th day, reaching approximately 5% of the population (S1 Fig).

We evaluated the metrics of interest over 1,000 iterations for each combination of scenarios and noise. We observed that when the epidemic is seeded in Queens and Kings, the epidemic size and the proportion of counties with at least one case are higher compared to an outbreak seeded in smaller counties (Fig 1A). As metrics can exist on very different scales, we calculate the normalized distribution of bootstrapped metrics where a minimum amount of noise is

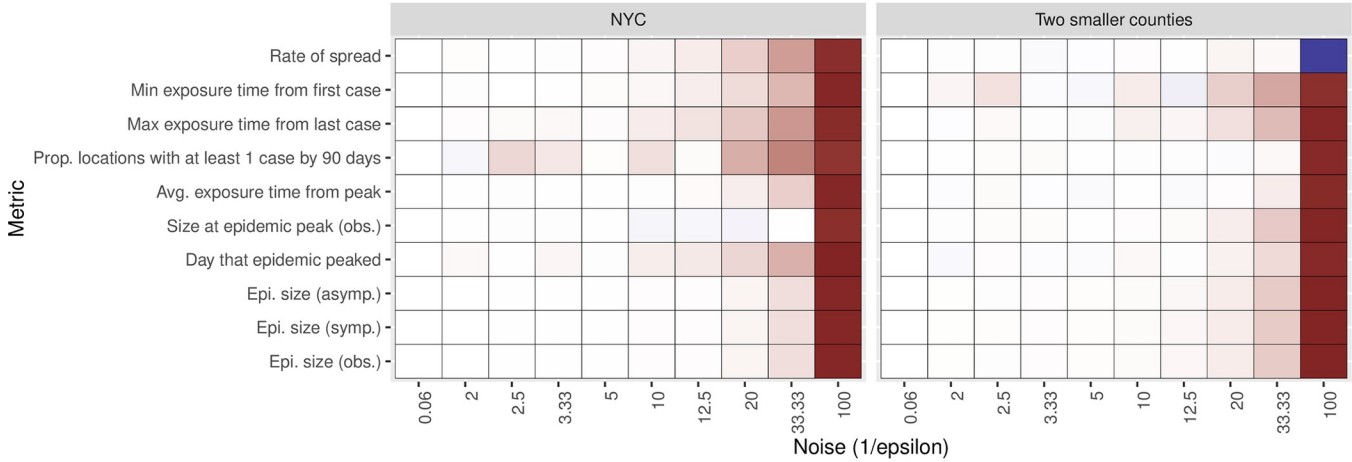

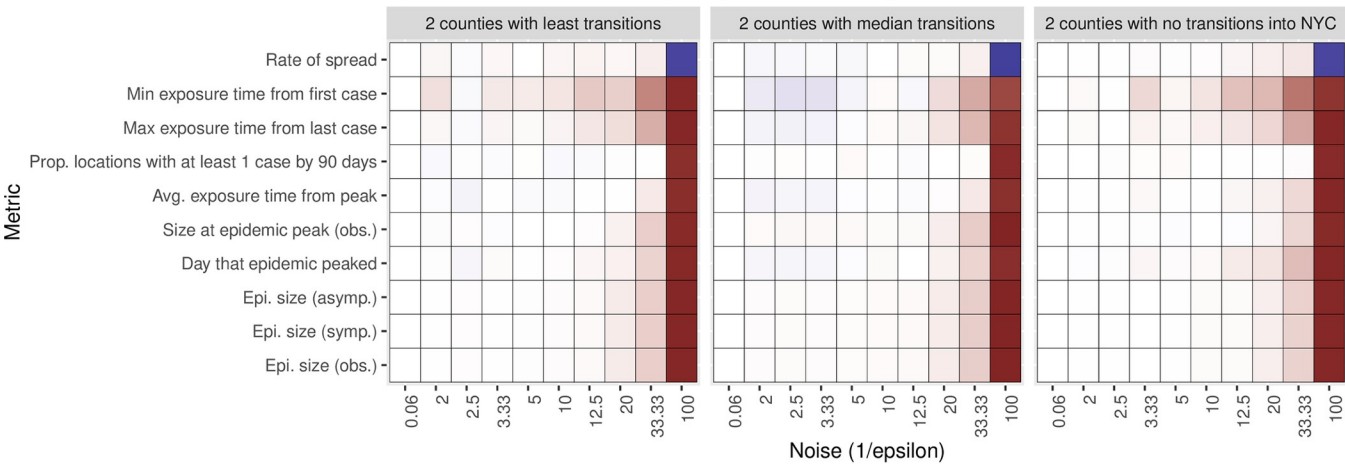

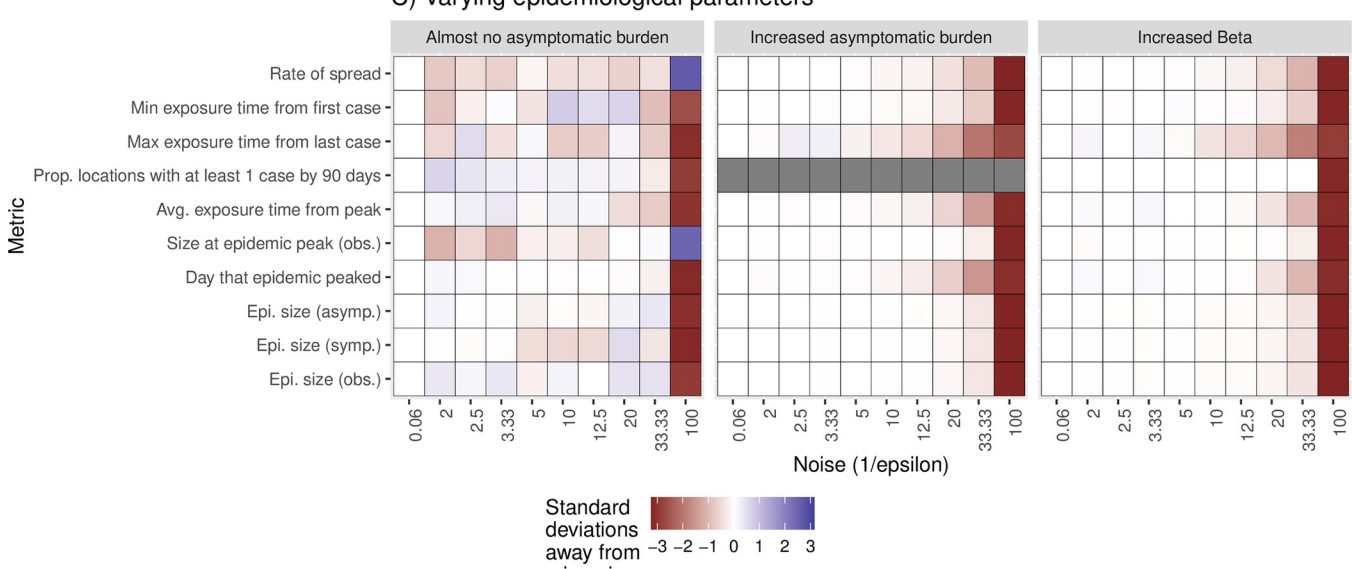

**Fig 1. Metapopulation metric distribution for different values of epsilon for Scenario A:** *Location of the first cases, B: change in the mobility network, and C: change in the parameters of the metapopulation model. As metrics can exist on very different scales, we calculate the normalized distribution of bootstrapped*

*metrics where a minimum amount of noise is added. We then compare this to the median value of bootstrapped values at increasing values of noise to describe the change from expectation.*

added. We then compare this to the median value of bootstrapped values at increasing values of noise to describe the change from expectation. When noise is above 20, the values for the epidemic size for observed, asymptomatic, and symptomatic infected, the size at the peak of the epidemic, and the proportion of counties with one case are lower than those obtained when the mobility matrix is not perturbed. However, the values obtained for the rate of spread, effective reproductive rate, risk of importation, probability of importation, and mean importation rate are higher than those obtained for the non-perturbed dataset (Fig 1).

## Scenarios with Epidemics in Well- and Poorly Connected Regions

To address how the effect of DP on network connectivity would impact predicted disease dynamics, we simulated an outbreak in three pairs of counties with varying levels of connectivity to Kings County. The first simulation in Monroe and Saratoga counties was designed to assess the impact of low connectivity (less than 20% of transitions during the period) on the disease dynamic. The second scenario targeted counties in the median of transitions, such as Putman and Westchester counties, to assess the dose-response effect of the epidemiological model. The third scenario was simulated in Schoharie and Lewis counties (no transition to Kings County during the period) to assess the impact in places that were isolated in the larger mobility network. When the outbreak is simulated in Monroe and Saratoga (Scenario 3), the epidemic begins around the 60th day and the number of infected persons reaches the maximum around the 150th day, with less than 1% of the total population living in this area infected. When the outbreak is seeded in medium connectivity areas such as Putnam and Westchester (Scenario 4), less than 0.6% of the population became infected around the 75th day after the epidemic peaks around the 40th day. When the outbreak is seeded in an area with low connectivity to Kings County, i.e., connectivity close to 0 such as Lewis and Schoharie (scenario 5), less than 0.07% of the population is infected around the 200th day since the epidemic only starts around the 90th day (S1 Fig).

We found that regardless of network connectivity, epidemiological metrics degraded as noise increased (Fig 1B). As such in the three scenarios addressing the change in the network of mobility, namely when i) the epidemic is sparked in two random counties having less than 20% transition to Kings County, ii) the epidemic is sparked in two random counties with a median transition to Kings County, and iii) the epidemic is sparked in a county with no transition to Kings County; we observed a similar pattern in the distribution of the metric to what we observed when there was an outbreak in small counties (scenario 2). Specifically, the size of the epidemic, the day that the epidemic peaks, the fraction of counties with at least one case, the size of the epidemic, the average exposure time, the maximum exposure time, and the minimum exposure time are smaller than the baseline. The spread rate, the effective reproductive rate, the importation risk, the mean importation risk rate, and the probability of infection are higher than the baseline, especially when the noise is above 33.33 (Fig 1B). We observed a significant change in the epidemiological metrics only when the value of noise added to perturb the transition matrix is above those of the scenario targeting the location of the first cases (small versus large county) (Fig 1B).

## Scenarios with varying epidemiological parameters

To address the nature of the epidemic, we simulated three changes in the trajectory of the epidemic in Kings and Queens counties. Specifically, we simulated i) a faster epidemic through

the increase of the transmission rate, ii) a heavy load of asymptomatic individuals, and ii) an absence of asymptomatic individuals in the population. When the transmission rate increases (scenario 6) we can observe that the epidemic starts around the 40th day and reaches its peak around the 75th with almost 3% of the population infected. When the fraction of symptomatic individuals increases, the size of the epidemic also increases and reaches 1.5% of the population around the 75th day since the epidemic starts around the 40th day after the first case (scenario 7). When the fraction of documented infection decreases (scenario 8), there is no declared epidemic, as only asymptomatic people are recorded in the population, reaching a fraction of 0.008% after the 100th day.

When the transmission rate increases, the epidemic spreads quickly (Fig 1C). When the asymptomatic rate increases, the probability of infection will subsequently increase. The trajectory of the epidemic is similar to the non-perturbed dataset. However, above the noise of 33.33, epidemiological metrics are either more conservative (lower than those of the baseline) or more volatile (higher than those of the baseline) (Fig 1C). Furthermore, we found that the fraction of counties with at least one case is not affected by the change in i) the transmission rate and ii) the fraction of symptomatic individuals (Fig 1C).

## Discussion

Several metapopulation models were developed throughout the SARS-CoV2 pandemic to inform decision making, predict the trajectory of the disease and identify weaknesses in the healthcare system [22–25]. The mobility data used to parameterize these models provided information on geographic and behavioral heterogeneity between populations, but these data could theoretically be used to identify individuals or their unique travel behavior, which warrants privacy preservation measures [26]. Our study shows that in metapopulation models that use mobility data, the application of privacy-preserving noise results in unbiased estimates of metrics of interest at a wide range of noise values with an upper limit that allows for a significant privacy-preserving budget.

We found that mobility matrices that are infused with noise values below 20, that is, loss of privacy loss of at least $\epsilon = 0.05$ per matrix, can help protect the privacy of individuals who contribute their data, while limiting bias in the estimation of public health measures of interest when used for epidemiological modeling. Importantly, as we have already added a minimum amount of noise to preserve privacy this limit represents a minimum threshold allowing for the addition of larger amount of privacy-preserving noise than previous studies have shown. Intuitively, adding noise to these mobility matrices may result in newly created connections between locations that would not otherwise be connected, strengthening connections that would otherwise be weak, or vice versa. In some cases, we may even see the removal of connections on specific days. Predictions of the spread of the rural area may be more affected than those of the areas connected to urban centers. However, sensitivity analyses could be performed to provide robustness, and the purpose and geographic scope of the model will dictate how important this degradation is.

As noise increases above 20, estimates such as the epidemic size, the day that the epidemic peaks, and the average epidemic size are biased downwards as the mobility matrix decreases connectivity to large population centers and distributes the epidemic into many smaller locations with lower contact rates. Similarly, estimates such as the rate of spread, the risk of importation, and the effective reproduction rate are biased upwards as mobility between smaller and poorly connected locations increases, leading to greater importation into areas with smaller population sizes. Our study demonstrates that for epidemiological metapopulation models

using mobility data, metrics estimates are fairly unbiased up to a noise threshold of 20, which provides greater privacy protection than previous studies [25, 27].

Although our pipeline only evaluated a specific combination of mobility data, metapopulation model, and metrics, it provides a " *plug-and-play* " interface for researchers to assess bias using proprietary models and mobility data [28]. As mobility data sets become increasingly available and used in metapopulation models, we provide a flexible framework to identify the evaluation-specific maximum privacy-preserving noise that can be incorporated into these mobility data before they result in biased outputs. Data providers can interact with researchers in many ways and the goal of this study is not to systematize this relationship. Instead, this "plug-and-play" framework can be used by researchers to simulate the effects of the application of differential privacy methods on their epidemiological parameters of interest. This would allow researchers to have an informed discussion with data providers before the data are sourced to identify an optimal threshold of noise which protects user privacy while also allowing for unbiased estimates of epidemiological parameters to be inferred.

## Methods

### Ethical statements

This study is a nonhuman subject research and does not necessitate neither an IRB approval nor an informed consent. Furthermore, all methods were performed in accordance with relevant guidelines and regulations.

### Study workflow

The pipeline workflow for the next analysis is represented in the following schematic architecture (Fig 2). This flow diagram shows the preprocessing before and after acquisition of the mobility data, and, most importantly, how synthetic data has been used to parameterize the metapopulation mode. Since obtaining non-processed data from third parties was impossible, we overlaid noise on pre-processed mobility data to determine the impact of differential privacy on the metapopulation model.

### Mobility data

We obtained mobility data from Camber Systems (the provider), a third-party analytics company that purchased advertising technology (ad tech) data from many data brokers. The data covered 90 days from August 15 to November 15, 2020, representing between 3–7% of the total American Community Survey (ACS), a county-specific population in New York State. The original data consisted of a log of user global positioning system (GPS) coordinates, sorted and grouped by a unique device identification number. These data have all the identifying information removed, cleaned to remove duplicate entries or unrealistic usage, used to calculate device-specific modal locations, and aggregated at the county-level [29]. The key metric of interest used in these analyses was movement between counties in 8-hour increments. Movement was defined as the change in a device's location from time period t-1 to the location of the device at time t. To further guarantee anonymity, the provider used a predefined group of devices per area, removed data that represented small numbers of devices, and applied an initial layer of privacy noise to the data set to ensure that the basic privacy preservation mechanisms were in place before providing access to these data to researchers [30]. However, the data provider did not disclose the initial privacy method and the degree of noise applied to maintain individual and group privacy. We then added an additional layer of post production differential privacy (PPDP) (see next section) and aggregated it into 24-hour blocks of time

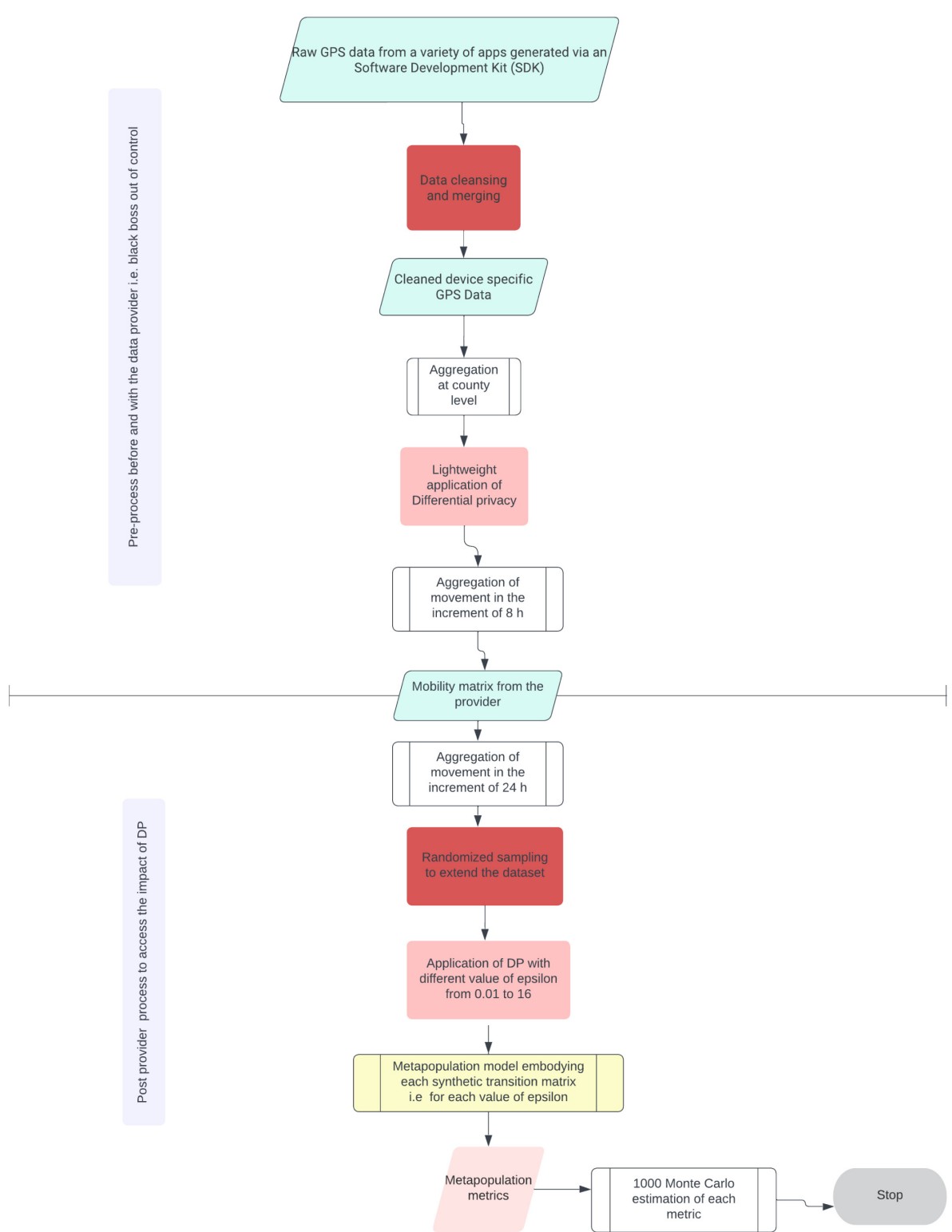

**Fig 2. Flow diagram showing the architecture of the modular software pipeline designed to quantify the tradeoff privacy utility of mobility data post-differential privacy processing in epidemiological models.** The boxes in pink represent the DP process, the boxes in red represent cleansing processes by both data providers and modelers, the boxes in green represent the data, the boxes in white are predefined processes, and the box in yellow stands for the metapopulation model.

with averaged transitions between counties. The process consists of generating an origin/destination matrix normalised to the ACS population for each county. The matrix was then randomly sampled and replicated 500 times to extend the data set time period. In most locations, the simulated outbreak was just beginning or at the exponential growth phase after the 90 days of the primary dataset. In light of this observation, it becomes particularly challenging to assess the impact of DP on epidemiological metrics which is why we used bootstrapping to form a mobility dataset of 500 days.

## Application of differential privacy

As background, a mechanism $M$ taking a database in a domain $H$ and producing outputs in a domain $R$ $M$: $H \rightarrow R$ is ($\epsilon$, $\delta$ ($\epsilon$, $\delta$)-differential private if and only if for every pair of neighboring databases $x$, $y \in H$, such that they differ in at most one entry, and for any subset of possible outputs $S \subseteq R$, we have

$$Pr[M(x) \in S] \leq e^{\epsilon} \, Pr[M(y) \in S] + \delta, \tag{1}$$

where the probability is taken over the randomness of the mechanism $M$ and is denoted the "security parameter' [31]. Eq (1) suggests that if two databases $x$, $y$ are sufficiently close due to the perturbation, then it becomes difficult for random attackers to uncover the privacy of the observed individuals. This is achieved by perturbing the true observations by adding noise from a carefully chosen distribution. The parameter quantifying the privacy loss $\epsilon$ represents the likelihood that an attacker with nearly full information about a database can determine whether their target is in the database. DP offers a quantifiable tradeoff between accuracy and privacy. Mobility data is aggregated data that could display the transmission of small groups of individuals. Our goal is to preserve the privacy of these groups and hide low transitions by applying differential privacy.

The Laplace mechanism is a common differential privacy mechanism, which adds Laplace noise to query values in which the noise scales with $\Delta/\epsilon$, where $\Delta$ is the query sensitivity. DP compositions adaptively allow us to design a mechanism with several building blocks, ensuring efficient privacy protection achievable using the advanced composition [10].

For all $\epsilon$, $\delta$, $\delta' > 0$, the class of ($\epsilon$, $\delta$)-differentially private mechanisms satisfies ($\epsilon'$, $k\delta+\delta'$)-differential privacy under $k$-fold adaptive composition for (Eq 2):

$$\epsilon\prime = \sqrt{2k log\left(\frac{1}{\delta\prime}\right)} \epsilon + k\epsilon(e^{\epsilon} - 1) \tag{2}$$

where $\delta'$ is a security parameter extremely small $\delta' = 2^{-30}$

To assess the tradeoff between accuracy and utility, we further privatize the synthetic data using the composition theorem with the privacy parameter epsilon ranging from 0.01 to 16 by the means of the Laplace mechanism using the 'smartnoise sdk' library [32]. The transition data contains the movements for 8-hour time blocks over 90 days, and using the advanced composition theorem with $k = 270$, the total privacy budget is as follows (Eq 3):

$$\epsilon\prime = \sqrt{540 \, log\left(\frac{1}{\delta\prime}\right)} \, \epsilon + 270 \, \epsilon(e^{\epsilon} - 1) = 84.6\epsilon + 270\epsilon(e^{\epsilon} - 1) \tag{3}$$

For $\epsilon = 0.01$, $\delta' = 2^{-30}$, we have $\epsilon' = 0.8731$ used to the existing deployment.

The rationale for using this range of epsilon lies in the fact that below 0.01 the infused noise is extremely large, compromising the accuracy of the transition matrix, and above 16 the total privacy budget is extremely large, compromising the privacy. More specifically, since the

transition matrix used already has privacy noise applied, with a value of $\epsilon = 16$ means, the synthetic transition obtained is similar to the one received from the provider. However, for $\epsilon = 0.01$, the synthetic data is more protective since low transitions are more hidden due to the large amount of noise added through the Laplace mechanism. To simplify interpretation, from here on, we evaluate *noise* which is the inverse of the privacy loss $\epsilon$.

## Metapopulation model

The disease dynamic was modeled with a Susceptible-Exposed-Infected Symptomatic-Infected asymptomatic model as follows (Eqs 4–7).

$$\frac{dS_i}{dt} = -\frac{\beta S_i I_i^r}{N_i} - \frac{\mu \beta S_i I_i^u}{N_i} \tag{4}$$

$$\frac{dE_i}{dt} = \frac{\beta S_i I_i^r}{N_i} + \frac{\mu \beta S_i I_i^u}{N_i} - \frac{E_i}{Z} \tag{5}$$

$$\frac{dI_i^s}{dt} = \alpha \frac{E_i}{Z} - \frac{I_i^s}{D} \tag{6}$$

$$\frac{dI_i^a}{dt} = (1 - \alpha)\frac{E_i}{Z} - \frac{I_i^a}{D} \tag{7}$$

where $S_i, E_i, I_i^s, I_i^a$ are the susceptible, exposed, infected symptomatic, infected asymptomatic, and total population in a county $i$.

The synthetic mobility datasets were integrated into the previous system (Eqs 4–7) and documented [33] by the following equations (Eqs 8–12),

$$\frac{dS_i}{dt} = -\frac{\beta S_i I_i^r}{N_i} - \frac{\mu \beta S_i I_i^u}{N_i} + \theta \sum_j \frac{M_{ij} S_j}{N_j - I_j^r} - \theta \sum_j \frac{M_{ji} S_i}{N_j - I_j^r} \tag{8}$$

$$\frac{dE_i}{dt} = \frac{\beta S_i I_i^r}{N_i} + \frac{\mu \beta S_i I_i^u}{N_i} - \frac{E_i}{Z} + \theta \sum_j \frac{M_{ij} E_j}{N_j - I_j^r} - \theta \sum_j \frac{M_{ji} E_i}{N_j - I_j^r} \tag{9}$$

$$\frac{dI_i^r}{dt} = \alpha \frac{E_i}{Z} - \frac{I_i^r}{D} \tag{10}$$

$$\frac{dI_i^u}{dt} = (1 - \alpha)\frac{E_i}{Z} - \frac{I_i^u}{D} + \theta \sum_j \frac{M_{ij} I_j^u}{N_j - I_j^r} - \theta \sum_j \frac{M_{ji} I_i^u}{N_j - I_j^r} \tag{11}$$

$$N_i = N_i + \theta \sum_j M_{ij} - \theta \sum_j M_{ji} \tag{12}$$

where $S_i, E_i, I_i^r, I_i^u$ are the susceptible, exposed, documented infected, undocumented infected, and total population in a county $i$.

The system of equations (Eqs 8–12) thus took into account both the mobility and the contagion describing the epidemic's evolution on the metapopulations network. We assumed that the randomness in the contagion followed a Poisson distribution and was documented elsewhere [33]. Most specifically, we seeded cases in a specific location, then, for each time $t$, the disease spread through the metapopulation network according to the transition matrix when

**Table 1. Parameters of the metapopulation model.**

| Parameters | Definition | Value | References |
|---|---|---|---|
| $\beta$ | Transmission rate | 0.8 | Estimated |
| $\mu$ | Factor of reduction of the transmission rate | 0.5 | Li et al. [33] |
| $\alpha$ | Fraction of symptomatic | 0.65 | Li et al. [33] |
| $Z$ | Average latency period | 3.6 | Li et al. [33] |
| $D$ | Average duration of infection | 3.14 | Li et al. [33] |
| $M_{ij}$ | Number of people travelling from county $j$ to county $i$ daily | - | Estimated |
| $M_{ji}$ | Number of people travelling from county $i$ to county $j$ daily | - | Estimated |
| $\theta$ | Multiplicative Travel Factor | 1 | Li et al. [33] |

people are moving between counties from the first day to the 500$^{\text{th}}$ day. Key parameters and sources in literature are described in Table 1.

## Epidemiological metrics

In reviewing epidemiological models using mobility data, we identified salient metrics of interest, including:

Probability of infection [34]: let denote $G(V, E)$ the mobility network of unknown topology where the vertices ($V$) are county/individuals with edges $(u, v)$ $E$. $u$ and $v$ are the contacts likely to result in infection. The model of disease dynamics can have four states described above (Eq 8:12) and is assigned to each individual. There is a probability that an epidemic will evolve through a particular sequence of states $\phi_1, \phi_2, \ldots, \phi_n$ and a probability $P$ that it will arrive at a certain state. A given state's $\phi_1$ probability is influenced by its previous state $\phi_0$ (Markov property). The probability of u is infected $\{I^u, I^r\}$ at the nth is given by (Eq 13):

$$P(u|\phi_1) = \sum\nolimits_{j=1}^{n} P(\phi_j(u) = \{I^u, I^r\}|\phi_1) \tag{13}$$

The risk of importation [35]: Let $F_i$ be the cumulative distribution function that the disease is likely imported to a county $j$ from a county $i$, $T_i$ be the probability associated with the travel from $i$, and $n_j$ the travel flux from $i$, the daily risk of importation $R_j$ is given by (Eq 14):

$$R_{j,t} = \frac{\sum_i F_i n_i T_i}{\sum_j F_j n_j} \tag{14}$$

Incubation period: It is the period of time between exposure to the disease-causing agent and the onset of symptoms [36].

Mean importation rate: is the average number of infected cases that move to $j$ during the epidemiological season (Eq 15)

$$\bar{R}_{j,t} = \frac{1}{N} \sum_i R_{j,t} \tag{15}$$

The effective reproduction rate [33] $R_e$: is a time-depend metric that measures how fast a disease is infectious given by the largest eigenvalue of the next-generation matrix method and is given by (Eq 16)

$$R_e = \alpha\beta D + (1-\alpha)\mu\beta D \tag{16}$$

Epidemic peak is the maximum number of infected over a time span of the epidemic [37]. Timing of the peak corresponds to the day the epidemic peak is reached [38].

**Table 2. Overview of the scenarios and the investigated question.**

|  | Characterisation of the Scenario | Scenario | Investigated question |
|---|---|---|---|
| 1 | Location of the first cases | The epidemic started in two large counties, i.e., Kings and Queens. | How does the perturbation of the transition matrix affect the epidemic curve when the epidemic starts in well-visited areas that are New York State? |
| 2 | | The epidemic started in two small counties, i.e., Allegany and Essex. | How does the perturbation of the transition matrix affect the epidemic curve when the epidemic starts in well-visited areas that are not New York State? |
| 3 | Change in Connectivity | The epidemic started in two counties with 20% of transitions, i.e., Saratoga and Monroe. | How does the perturbation of the transition matrix affect the epidemic curve when the epidemic started in a region with low connectivity to New York City? |
| 4 | | The epidemic started in two random counties in the median of transitions, i.e., Westchester and Putnam. | Is there a "dose-response" of the interaction between transitions? |
| 5 | | The epidemic started in the counties of Schoharie and Lewis with no transitions to New York City, | How does the perturbation of the transition matrix affect the epidemic curve when the epidemic starts in locations that seem to be isolated from New York City? |
| 6 | Change in parameters of the metapopulation model | Increase the contact rate ($\beta = 0.9$) | How does the perturbation of the transition matrix affect the epidemic curve when the epidemic is more transmissible? |
| 7 | | Increase asymptomatic burden ($\alpha = 0.75$) | How does the perturbation of the transition matrix affect the epidemic curve when the burden of transmission is passed on by asymptomatic individuals? |
| 8 | | Decrease the asymptomatic burden ($\alpha = 0.01$) | How does the perturbation of the transition matrix affect the epidemic curve when only symptomatic individuals transmit? |

The rate of spread represents the ratio of infected by susceptible over a time-span [39].

Proportion of counties with at least one case [40].

Epidemic size is the total number of infected divided by the population size of each county and then multiplied by 100,000 individuals [41].

Size at the epidemic peak is the previous ratio by the day of the epidemic peak [42].

## Epidemiological scenarios

To assess the effect of noise on these metrics, we evaluated eight scenarios with three salient characteristics and provided a general formula to incorporate more. We evaluated scenarios where the epidemiological metrics of interest were driven by i) the location of the first case, ii) changes in connectivity, and iii) changes in epidemiological parameters (Table 2).

We explored several spatial epidemiological questions (Table 2) with our scenarios, including how the place of outbreak affects the dynamics of the disease, how connectivity networks could potentially affect epidemic dynamics, and how DP might ultimately affect the metric we are interested in.

To assess the impact of privacy on the epidemiological metric, we ran each set of parameters through 1000 Monte Carlo iterations and visualised the results.

## Supporting information

**S1 Fig. Simulated scenarios epidemiological curve embedding perturbed mobility matrices.** Scenarios 1 and 2 the disease is spread in two large and small counties, respectively; Scenarios 3, 4, and 5 the epidemic occurred in counties with no, medium to high connectivity with neighboring counties; Scenarios 6, 7, and 8 key parameters such as the burden of asymptomatic, the contact rate varied. $I^s$, $I^a$, and Obs are infected symptomatic, infected asymptomatic, and observed.
(DOCX)

## Author Contributions

**Conceptualization:** Navin Vembar, Satchit Balsari, Caroline O. Buckee, Salil Vadhan, Nishant Kishore.

**Data curation:** Merveille Koissi Savi, Akash Yadav, Wanrong Zhang, Navin Vembar, Nishant Kishore.

**Formal analysis:** Merveille Koissi Savi, Akash Yadav, Wanrong Zhang, Nishant Kishore.

**Funding acquisition:** Navin Vembar, Andrew Schroeder, Satchit Balsari, Caroline O. Buckee, Salil Vadhan.

**Investigation:** Satchit Balsari, Caroline O. Buckee, Salil Vadhan.

**Methodology:** Merveille Koissi Savi, Akash Yadav, Wanrong Zhang, Nishant Kishore.

**Project administration:** Satchit Balsari.

**Resources:** Merveille Koissi Savi.

**Software:** Merveille Koissi Savi, Akash Yadav, Nishant Kishore.

**Supervision:** Navin Vembar, Andrew Schroeder, Satchit Balsari, Salil Vadhan, Nishant Kishore.

**Validation:** Andrew Schroeder, Caroline O. Buckee, Salil Vadhan, Nishant Kishore.

**Visualization:** Merveille Koissi Savi, Navin Vembar, Nishant Kishore.

**Writing – original draft:** Merveille Koissi Savi.

**Writing – review & editing:** Merveille Koissi Savi, Akash Yadav, Wanrong Zhang, Navin Vembar, Andrew Schroeder, Satchit Balsari, Caroline O. Buckee, Salil Vadhan, Nishant Kishore.

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
