## [Decision Letter · Decision Letter 0]

9 May 2023

PDIG-D-23-00098

A standardised differential privacy framework for epidemiological modelling with mobile phone data

PLOS Digital Health

Dear Dr. Kishore,

Thank you for submitting your manuscript to PLOS Digital Health. After careful consideration, we feel that it has merit but does not fully meet PLOS Digital Health's publication criteria as it currently stands. Therefore, we invite you to submit a revised version of the manuscript that addresses the points raised during the review process.

Please submit your revised manuscript within 60 days Jul 08 2023 11:59PM. If you will need more time than this to complete your revisions, please reply to this message or contact the journal office at digitalhealth@plos.org. We look forward to receiving your revised manuscript.

Kind regards,

Michele Tizzoni

Academic Editor

PLOS Digital Health

Journal Requirements:

1. We ask that a manuscript source file is provided at Revision. Please upload your manuscript file as a .doc, .docx, .rtf or .tex.

2. Please provide separate figure files in .tif or .eps format.

Additional Editor Comments (if provided):

Both referees have found the manuscript to be a valuable work that addresses a timely topic of interest. 

Please, in your revision, carefully consider the requests of both referees especially regarding the clarity of your methods and the description of the epidemiological scenarios.

Reviewers' comments:

Reviewer's Responses to Questions

**Comments to the Author**

1. Does this manuscript meet PLOS Digital Health’s publication criteria? Is the manuscript technically sound, and do the data support the conclusions? The manuscript must describe methodologically and ethically rigorous research with conclusions that are appropriately drawn based on the data presented.

Reviewer #1: Yes

Reviewer #2: Partly

2. Has the statistical analysis been performed appropriately and rigorously?

Reviewer #1: I don't know

Reviewer #2: Yes

3. Have the authors made all data underlying the findings in their manuscript fully available (please refer to the Data Availability Statement at the start of the manuscript PDF file)?

Reviewer #1: Yes

Reviewer #2: Yes

4. Is the manuscript presented in an intelligible fashion and written in standard English?

Reviewer #1: Yes

Reviewer #2: Yes

5. Review Comments to the Author

Reviewer #1: This paper addresses an important important topic, and presents a timely application of differential privacy. However, I believe that the presentation needs to be improved before publication.

Specifically, while I believe the privacy methodology to be sound, I have concerns about its presentation:

1. The exact mechanism used is never explicitly described, and as such it is hard to ensure that it satisfies differential privacy (especially in regards to the computation of the sensitivity). My understanding is that the original data already comes in aggregated form (transition matrices for 8-hour slots), to which individual users contribute at most once (sensitivity = 1). Noise is then added to these matrices before they are aggregated to form daily transition matrices. Is that accurate?

2. In line 233, you mention that the data already has an "initial layer of privacy noise". This should be explained in more detail: what mechanism was used? Is it DP? If so, what privacy budget was used? 

3. The computation of epsilon on line 265 is inaccurate, but I suspect this is due to a typo in the value of delta. It seems you used 1.75e-6, which is reasonable. It would also be good to explain briefly how you chose this value of delta.

4. In lines 31 and 44, the paper mentions that an "epsilon of 0.05" could be used: this is a bit misleading due to the composition, and the total value (around 4 after advanced composition) should be used instead in these segments.

I found the epidemiological analysis interesting (although this is outside my area of expertise), but I think the results could be presented in a clearer way. At the moment, it is sometimes unclear whether the focus is on the results themselves or the impact of the privacy noise on the results. If possible, it would be good to show, e.g., a plot displaying how DP impacts key metrics. Your recommendation in the asbtract of using epsilon=0.05 should also be more explicit in the results section.

Minor comments: 

- Could you please explain why and how you resample the data (line 238), and what impact (if any) it would have on the analysis?

- Your definition of differential privacy (lines 244-245) should include delta, as you use it later for advanced composition.

- I was a bit surprised by some of the references cited for the privacy side. In line 73, if you wish to say that aggregated data can be re-identified, I would cite https://arxiv.org/abs/1708.06145 rather than [8]. In line 75, I think references [9-15] do not represent "several privacy frameworks", at least as would be understood by the computer science community (i.e., k-anonymity, l-diversity, federated learning). Similarly, reference [18] on line 90 points to the OpenDP website, which does not support the claim. For that sentence, it would also be good to add references for the use of DP by Google etc. (see, e.g., https://desfontain.es/privacy/real-world-differential-privacy.html for a list of up-to-date references)

Reviewer #2: The authors present an interesting paper on the impact of differential privacy techniques applied to mobility data on the output of epidemiological models. I believe this paper is very relevant. Given the central role that mobility data have acquired during the past years in computational epidemiology, similar contributions are needed as we build better data and modeling preparedness for the next global health emergency. 

I do not have major concerns with the validity of the work, but I have some remarks mostly aimed at improving the accessibility of the paper: 

1) It was not straightforward for me to understand what Fig 1 is showing and connect this with the commentary in the Results section. I would recommend to include a caption to Fig.1 and a brief description of it at the beginning of the results section. For example, apart from the legend in the figure it is never mentioned what the colors of the heatmap in Fig. 1 indicate and how they should be interpreted.

2) The epidemiological metrics need a better explanation. Currently, they are just listed along with references in the methods section. While the interpretation of some metrics is quite self-explanatory, for others it may not be as straightforward especially for readers that are less familiar with epidemiological modeling. Therefore, I recommend to include more details on the definition of these metrics in the Methods section.

3) I have similar remark also for the 8 epidemiological scenarios considered. At the current stage, the methods section has a very brief paragraph related to their description. I believe the paper would benefit from more details about the simulation scenarios. 

4) I would like to see clarified a bit more the sampling procedure of the mobility matrix. It is my understanding that the simulations last for 500 days, while the mobility data covers roughly 4 months. It is unclear to me at the moment how we go from a matrix M_t that varies daily for ~4 months to the 500 days one. 

5) The providers of mobility data apply privacy filters to ensure that the privacy of users is preserved. Therefore, researchers receive preprocessed data that has undergone privacy-preserving algorithms. This is also the case with the data used in the paper under consideration. As the authors mentioned, the data has undergone privacy-preserving algorithms twice. However, I have two questions concerning this point.

5a) First, I wonder if this can bias the maximum estimated threshold of noise that can be applied without biasing too much the epidemiological metrics. My point is that it may be higher than 20 as the data have already some noise. While I do not think authors should change their analysis, this point should be at least discussed.

5b) My second point concerns the authors' approach to operationalizing the proposed methodology. In its current form, the paper lacks a comprehensive discussion on this topic. Mobility data is typically provided with noise, and researchers have limited knowledge about the methods used by data providers to ensure user privacy. Therefore, it is unclear how the proposed methodologies can overcome these challenges. One potential solution is to collaborate with data providers and encourage them to apply privacy-preserving algorithms that do not compromise the accuracy of epidemic models. Alternatively, privacy-preserving algorithms can be reapplied to the data, but this approach requires more caution as different providers may have applied vastly different preprocessing techniques. Overall, the paper would benefit from a more in-depth exploration of the potential challenges associated with operationalizing the proposed methodology and a thorough discussion of possible solutions.

6. PLOS authors have the option to publish the peer review history of their article (what does this mean?). If published, this will include your full peer review and any attached files.

**Do you want your identity to be public for this peer review?** For information about this choice, including consent withdrawal, please see our Privacy Policy.

Reviewer #1: No

Reviewer #2: No

---

## [Decision Letter · Decision Letter 1]

3 Sep 2023

A standardised differential privacy framework for epidemiological modelling with mobile phone data

PDIG-D-23-00098R1

Dear Dr. Kishore,

We are pleased to inform you that your manuscript 'A standardised differential privacy framework for epidemiological modelling with mobile phone data' has been provisionally accepted for publication in PLOS Digital Health.

Best regards,

Michele Tizzoni

Academic Editor

PLOS Digital Health

Reviewer Comments (if any, and for reference):

Reviewer's Responses to Questions

**Comments to the Author**

1. If the authors have adequately addressed your comments raised in a previous round of review and you feel that this manuscript is now acceptable for publication, you may indicate that here to bypass the “Comments to the Author” section, enter your conflict of interest statement in the “Confidential to Editor” section, and submit your "Accept" recommendation.

Reviewer #1: All comments have been addressed

Reviewer #2: All comments have been addressed

2. Does this manuscript meet PLOS Digital Health’s publication criteria? Is the manuscript technically sound, and do the data support the conclusions? The manuscript must describe methodologically and ethically rigorous research with conclusions that are appropriately drawn based on the data presented.

Reviewer #1: Yes

Reviewer #2: Yes

3. Has the statistical analysis been performed appropriately and rigorously?

Reviewer #1: Yes

Reviewer #2: Yes

4. Have the authors made all data underlying the findings in their manuscript fully available (please refer to the Data Availability Statement at the start of the manuscript PDF file)?

Reviewer #1: Yes

Reviewer #2: Yes

5. Is the manuscript presented in an intelligible fashion and written in standard English?

Reviewer #1: Yes

Reviewer #2: Yes

6. Review Comments to the Author

Reviewer #1: (No Response)

Reviewer #2: (No Response)

7. PLOS authors have the option to publish the peer review history of their article (what does this mean?). If published, this will include your full peer review and any attached files.

**Do you want your identity to be public for this peer review?** For information about this choice, including consent withdrawal, please see our Privacy Policy.

Reviewer #1: None

Reviewer #2: None
